# Salivary Cytokines as Biomarkers for Oral Squamous Cell Carcinoma: A Systematic Review

**DOI:** 10.3390/ijms22136795

**Published:** 2021-06-24

**Authors:** Elena Ferrari, Margherita E. Pezzi, Diana Cassi, Thelma A. Pertinhez, Alberto Spisni, Marco Meleti

**Affiliations:** 1Department of Medicine and Surgery, University of Parma, 43125 Parma, Italy; elena.ferrari@unipr.it (E.F.); alberto.spisni@unipr.it (A.S.); marco.meleti@unipr.it (M.M.); 2Centro Universitario di Odontoiatria, University of Parma, 43125 Parma, Italy; margherita.pezzi@gmail.com; 3Dentistry and Oral and Maxillofacial Surgery, Department of Surgical, Medical, Dental and Morphological Science with Interest in Transplant Oncological and Regenerative Medicine, University of Modena and Reggio Emilia, 40100 Modena, Italy; diana.cassi@unimore.it

**Keywords:** oral cancer, oral squamous cell carcinoma, biomarkers, cytokines, oral potentially malignant disorders

## Abstract

The prognosis of patients with oral squamous carcinoma (OSCC) largely depends on the stage at diagnosis, the 5-year survival rate being approximately 30% for advanced tumors. Early diagnosis, including the detection of lesions at risk for malignant transformation, is crucial for limiting the need for extensive surgery and for improving disease-free survival. Saliva has gained popularity as a readily available source of biomarkers (including cytokines) useful for diagnosing specific oral and systemic conditions. Particularly, the close interaction between oral dysplastic/neoplastic cells and saliva makes such fluid an ideal candidate for the development of non-invasive and highly accurate diagnostic tests. The present review has been designed to answer the question: “Is there evidence to support the role of specific salivary cytokines in the diagnosis of OSCC?” We retrieved 27 observational studies satisfying the inclusion and exclusion criteria. Among the most frequent cytokines investigated as candidates for OSCC biomarkers, IL-6, IL-8, TNF-α are present at higher concentration in the saliva of OSCC patients than in healthy controls and may therefore serve as basis for the development of rapid tests for early diagnosis of oral cancer.

## 1. Introduction

Oral cancers are among the most common malignant tumors worldwide, with 354,900 new cases and 177,400 deaths reported in 2018 [1]. More than 90% of all oral malignancies are oral squamous cell carcinomas (OSCC) [2]. While in Western countries, OSCC accounts for about 4% of all cancers; in India and Southeast Asia, it reaches up to 40% [3].

The clinical course of the disease is strictly related to the time of diagnosis [4]. According to the Surveillance, Epidemiology, and End Results (SEER) Program (based on the U. S. population), when OSCC is confined to the primary site, the 5-years relative survival rate is about 85.1%, decreasing to 66.8% and to 40.1% for tumors spread to regional lymph nodes and to distant sites, respectively [5].

Classical risk factors for OSCC development in Western countries are smoking and alcohol abuse. The role of some human papillomaviruses (HPV) infection has been clearly established for carcinomas of the oropharynx, but doubts still exist on the oncogenic potential of such viruses in the rest of the oral cavity [6].

As for the rest of the head and neck tumors, the pathogenesis of OSCC is still a matter of controversy. Most authors agree on a multi-step process in which the accumulation of genetic and epigenetic changes affect protein expression, thus altering a variety of signaling pathways [7]. Gene alterations may range from the gain or loss of single nucleotides, up to partial or complete deletion of chromosomes. The loss of genes has been described in significant conjunction with inactivating mutations of tumor suppressor genes [8].

In a recent review on the molecular landscape in head and neck cancer, Leemans et al. identified five cellular processes that may be dysregulated in OSCC pathogenesis: cell cycle, growth signal, survival, WNT signaling and epigenetic regulation [9].

Diagnosis of OSCC is based on clinical inspection followed by biopsy and histopathological evaluation of suspicious tissues. Vital staining (e.g., toluidine blue) and auto-fluorescence imaging may highlight tissues undergoing rapid cell division and represent complementary diagnostic aids, as they can differentiate normal from dysplastic or cancerous tissues and direct the biopsy to the target site [10]. Radiographic imaging is used to investigate the involvement of the surrounding tissue and structures, such as muscles, bone, and lymph nodes; in particular, computed tomography (CT) and magnetic resonance imaging (MRI) are essential for bone and neck nodes evaluation [11].

Some OSCCs are preceded by oral potentially malignant disorders (OPMDs), including leukoplakia, erythroplakia, lichen planus and oral submucous fibrosis [12,13], at the histological level.

Visual examination alone, however, may lead to ignoring subtle lesions and failing to differentiate malignant from benign oral conditions. As a result, OSCC is diagnosed at an already advanced stage, adversely affecting the survival rate. Many studies are, therefore, focused on the identification of easily accessible early diagnostic biomarkers.

### 1.1. Salivary Biomarkers for OSCC

In the last decade, studies have focused on the analysis of body fluids, otherwise named “liquid biopsy”, for the detection of OSCC diagnostic and/or prognostic biomarkers [14]. Saliva has raised considerable interest [15,16,17] due to its proximity to cancer cells, accessibility, non-invasive collection, and cost-effective sampling. Progress in the comprehension of OSCC development at the molecular level has favored the identification, in unstimulated whole saliva, of several potential biomarkers reporting on proteomic or metabolic activity, as well as on genomic and epigenetic alterations of malignant cells [18,19], thus favoring the early detection and diagnosis of a possible pathological state [16,20].

Although standardized protocols for reliable biomarkers are still under development, recent data suggest that the inclusion of “salivary liquid biopsy” has great potential for OSCC diagnosis and management, as it improves prognosis, therapy, and follow-up [20,21].

### 1.2. Cytokines and Cancer

Among proteinaceous salivary biomarkers, studies on the molecular role of cytokines in the tumor microenvironment (TME) revealed that they are involved in processes leading to the initiation, growth, invasion, and metastasis of cancer [22].

TME is a complex tissue that significantly deviates from original tissue homeostasis and contains, in addition to neoplastic cells, a network of stromal cells including fibroblasts, vascular cells, and immuno-inflammatory cells [23,24]. The latter include innate immune cells (macrophages, neutrophils, mast cells, myeloid-derived suppressor cells, dendritic cells, natural killer cells) and adaptive immune cells (T and B lymphocytes), suggesting that both inflammation and immune response play a critical role in tumorigenesis.

The inflammatory response, orchestrating a complex interplay between stromal and cancer cells, is now recognized to affect different aspects of tumor development and progression. In fact, under the influence of inflammation, neoplastic and stromal cells interact and control the tumor evolution producing cytokines, growth factors, proangiogenic factors, and remodeling enzymes of the extracellular matrix [22,25].

Cytokines present in TME are low-molecular weight proteins produced by immune and stromal cells that control cell proliferation, survival, migration, as well as immune cell activation. They also modulate the anti-tumoral immune response; although, in chronic inflammation, they induce tumor transformation [22].

Pro-inflammatory cytokines, such as interleukin-1 (IL-1), IL-6, IL-8, INF-γ, and TNF-α, favor tumor growth and invasion by promoting cell proliferation, epithelial-mesenchymal transition and angiogenesis, while increasing tumor immune surveillance. By contrast, anti-inflammatory cytokines, such as IL-4, IL-10, and IL-13, counterbalance the proliferative potential of the inflammatory ones. Moreover, they are immunosuppressive agents that can enable tumor cells to evade immune surveillance, thus preventing tumor rejection [22,26,27,28].

The imbalance between local and systemic cytokine levels, which may promote tumor growth and progression, provides evidence of the relationship between chronic inflammation and oral cancer or potentially malignant diseases [27]. The pro-inflammatory and pro-angiogenic IL-1, IL-6, IL-8, and TNF-α are involved in OSCC development by promoting cell survival and proliferation. In fact, they up-regulate positive cell cycle controllers, such as nuclear factor kappa B (NF-κB), signal transducer and activator of transcription (STAT) proteins, and mitogen-activated protein kinase/extracellular signal-regulated kinase (MAPK/ERK) pathways [29].

A recent article [30] describes the molecular mechanisms underlying TNF-α-mediated OSCC invasion in vitro. The authors propose that oral chronic/acute inflammation leads to a “pro-tumor” phenotype by recruiting neutrophils that will establish a feedback loop with OSCC cancer cells. According to their interpretation, (1) TNF-α released by neutrophils during oral inflammation activates OSCC cells, inducing gene expression changes, responsible for invadopodia formation and invasion (by PI3K and Src activation), and for inflammatory cytokines/chemokines release; (2) OSCC-released cytokines and chemokines lead to the activation of nearby neutrophils; (3) further inflammatory signaling and TNF-α release from activated neutrophils complete the positive feedback loop with OSCC cells. It is worth noting that, among the most overexpressed genes in OSCC cells is IL-8, another pro-inflammatory cytokine, and the matrix metallopeptidase 9 (MMP9), an enzyme related to tumor invasion, is associated with basement membrane degradation and extracellular matrix remodeling. The proposed model may be considered paradigmatic of the interplay between cancer cells and immuno-inflammatory cells of the TME, mediated by cytokines expression.

In the wake of these findings, the identification of salivary cytokines for the early diagnosis of OSCC has been one of the primary targets of the research in the field of oral cancer. In the past 15 years, many studies focused on the determination of cytokine concentrations in saliva, suggesting that an abnormal cytokine level might play a role as an OSCC biomarker.

The present systematic review attempts to answer the question: “Is there evidence to support the role of specific salivary cytokines in the diagnosis of OSCC?”

## 2. Results

### 2.1. Study Selection and Characteristics

A search of various databases yielded 7479 unique articles (Figure 1). Upon title reading, 7337 articles were excluded because they were irrelevant to the review′s query. The remaining 142 articles (plus five additional articles retrieved from reference lists) qualified for further analysis (abstract evaluation) and 33 papers were finally selected for full-text evaluation. Among them, six were excluded based on one or more of the following features: inclusion of patients with a diagnosis of laryngeal adenocarcinoma or head and neck squamous cell carcinoma (HNSCC); the presence of data already included in another selected article; unavailability of the full text.

All the included studies are observational (cross-sectional or longitudinal). Some of them, when evaluating the cytokine level, considered different OSCC histological grades and clinical stages. Based on the methodological design, studies were divided into three groups: Group I (*n* = 10)—cross-sectional studies comparing salivary cytokine levels of patients with OSCC to control subjects; Group II (*n* = 12)—cross-sectional studies comparing salivary cytokine levels of patients with OSCC to control subjects and evaluating their relationship to OSCC histological grading and/or clinical staging; Group III (*n* = 5)—longitudinal studies considering tumor excision as a treatment and comparing salivary cytokine levels before and after surgery. Twelve studies from Groups I and II also included patients with OPMD.

Among all studies, only six showed a low risk of bias, since 25% of the answers to the JBI Critical Appraisal Checklist [31] were negative; the remaining were assessed as being at no risk of bias (Appendix A).

Study characteristics and cytokine level comparisons extracted from the three groups are listed in Table 1, Table 2 and Table 3.

### 2.2. Main Findings

The studies analyzed demonstrate that numerous cytokines (e.g., IL-6, IL-8, TNF-α) are present in the saliva of OSCC patients at a significantly different concentration when compared to healthy persons. Out of 27 studies, 25 highlight an increase of salivary cytokines’ concentration in OSCC patients. In the remaining two [52,53], the comparison with healthy subjects is not considered, since each OSCC patient serves as a control when comparing the concentrations before surgery vs. after surgery.

In seven studies of Group II [27,28,43,44,45,46,47], the cytokines salivary level increases regularly when moving gradually from well differentiated to poorly differentiated OSCC lesions. The trend is observed for IL-6, IL-8, TNF-α and IL-1RA, demonstrating that they can be associated with disease aggressiveness and severity. Patients with early stage OSCC (stage I/II or T1/T2) have higher concentrations of IL-6, IL-8, IL-1β, TNF-α, IFN-γ, MIP-1β, and Growth Regulated Oncogene (GRO) when compared to controls [27,42,48,49], unveiling a potential discriminatory power of the cytokines’ level to reveal early phase OSCC development.

In addition, several longitudinal studies based on surgical resection of OSCC show that, in the presence of OSCC, an increased concentration of cytokines IL-6, IL-8, IL-1β, IL-17, VEGF, MIP-1β and IP-10 is observed [52,53] as compared to the after-surgery level.

Finally, it is worth noting that, in nearly all studies, salivary IL-6, IL-8, TNF-α levels in OPMD patients, are lower than in OSCC patients and that, for all cytokines, their levels are significantly different with respect to the control population [27,29,32,34,35,41,43,44,45,46,47,48,50].

## 3. Discussion

It is well accepted that inflammation and cell-mediated immunity are active players in the control of oral carcinogenesis progression [56]. Cells’ evasion from immune surveillance has been described as a primary step in oncogenesis [57]. Pro-inflammatory cytokines (IL-6, IL-8, TNF-α, soluble interleukin-2 receptor (sIL-2R)) are key-molecules involved in the crosstalk between stromal and cancer cells, their expression being, to some extent, associated with tumor growth promotion or inhibition [58].

Serum levels of pro-inflammatory growth factors and cytokines in patients with OSCC or other oral potentially malignant disorders are still poorly investigated. Recent results would demonstrate that the serum level of IL-6, IL-8 and sIL-2R is significantly higher in patients with OSCC compared to healthy controls and to patients with OPMDs [56]. Thus, it appears feasible to accept that some salivary cytokines might be considered reliable biomarkers for OSCC diagnosis.

All the selected studies identify the institutional care facility that enrolled the OSCC patients and, in most cases, they mention the approval of an institutional ethical committee, thus presenting appropriate information about ethical protection.

As for saliva collection and processing, we ascertained, overall, a reasonable procedural consistency. In fact, most of the studies (1) analyze unstimulated whole saliva, obtained via passive drool; (2) include a preliminary step of centrifugation at 4 °C to separate cells and debris; (3) freeze the saliva supernatant, usually at −80 °C, until the analysis. Nevertheless, only in a few studies did the salivary protein content, including cytokine molecules, appear to be protected by the addition of protease inhibitors [28,33,36,40,48].

In all studies, an enzyme-linked immunosorbent assay (ELISA), based on antibodies against specific cytokines and a colorimetric revelation system, turned out to be the method of choice for cytokine quantitation. The method allows the accurate detection of the antigen, but measures only one cytokine in each sample, with consequent sample waste and high cost when the purpose is to measure multiple cytokines. In four studies that investigated the potential variation in the concentration of several cytokines, a multiplex bead-based immunoassay was preferred [27,28,42,52]. Besides the high throughput multiplex analysis, the major advantage is its broader dynamic range of cytokine concentration measurable as compared to the ELISA test.

Since the possible presence of dental and periodontal infection may influence salivary cytokine concentrations [59], many of the selected studies evaluate the periodontal status of the study participants. While most of them simply exclude subjects with periodontitis from control and/or patient groups or include control subjects with periodontal status matched with the patient group, the studies of Brailo and co-workers [29] and Sato and co-workers [54,55] take into account the periodontal status of all the study participants. Thus, by using the community periodontal index, as described by the World Health Organization (WHO), they can demonstrate that the salivary concentrations of IL-6 and TNF-α are not affected by periodontal health.

In addition, except for a few cases [33,34,36,39], the authors agree on excluding subjects with any relevant comorbidity, for example, chronic or acute illnesses, autoimmune diseases, chronic inflammatory diseases, pathological dry mouth syndrome, or the inability to collect a sufficient saliva sample. We believe that the application of this exclusion criterion constitutes good practice, consistent with the diagnostic purposes of this topic area.

The interfering behavioral habits considered are mainly alcohol consumption and tobacco chewing or smoking. The use of these substances is frequent in OSCC patients, although preferential habits appear to be associated with the specific geographical origin of the patients. In only nine studies [27,28,29,35,39,43,46,49,55], the authors tested whether the salivary cytokine level was affected by social habits. As far as IL-1α, IL-1β, IL-6, TNF-α and IL-8 are individually concerned, their salivary concentrations did not appear to be affected by (or correlated to) specific habits.

In thirteen articles of Groups I and II, salivary cytokine level is also assessed in the OPMD patient group and compared with OSCC and control groups (Table 1 and Table 2). OPMD patients suffer prevalently from leucoplakia, but also from oral sub mucous fibrosis or oral lichen planus, while erythroplakia did not appear in any study, possibly because of its rare occurrence. In Group II, the salivary levels of IL-6, IL-8, and TNF-α of OPMD patients show a statistically significant higher concentration when compared with the control, but lower when compared to OSCC patients [27,41,43,44,45,46,47,50]. Likewise, in Group I, comparable findings are obtained with IL-6 and TNF-α salivary concentrations [29,35]. In these cases, the salivary cytokines concentration allows us to differentiate between OPMD, OSCC and healthy subjects, thus reflecting a multi-diagnostic potential and corroborating the use of selected cytokines as biomarkers.

Among the studies including OPMD subjects, six articles investigated the diagnostic utility of the salivary cytokine level by ROC curve analysis and compared the ability of selected cytokines (IL-6, IL-8, TNF-α) to differentiate between patients with OSCC and OPMD [27,40,43,44,45,46]. The significant area under the curve (AUC) ranges from 0.70 to 0.99, reflecting an effective power of discrimination.

As observed in some studies [27,40,49], the combination of different salivary biomarkers is of great value for OSCC detection. A predictive model based on six cytokines and used for distinguishing OSCC from control subjects yielded a sensible increase in AUC as compared to individual cytokine analysis [27]. Additionally, the combination of cytokine proteomic and transcriptomic biomarkers generated an increased discriminatory effect between OSSC and control subjects [40,49], although further inclusion of a risk factor exposure (areca nut, smoking) provided the best panel of variables useful for OSCC detection [40].

Unfortunately, in OSCC patients, treatment outcome and prognosis are seriously affected not only by late diagnosis but also by frequent loco regional recurrences. Sato and co-workers [54,55] perceived the potential of IL-6 in terms of recurrence prediction and demonstrated that the sequential analysis of salivary cytokines post-treatment could be a useful marker for the diagnosis of early and late loco regional recurrence (Table 3), a common cause of mortality in patients with OSCC. To give one example, they measured an increased postoperative IL-6 concentration in subjects with early recurrence as compared to subjects without recurrence. In these two longitudinal studies, inevitably, a limited number of OSCC patients received pre-operative or post-operative treatment. Nevertheless, the authors point out that there are no statistical differences in salivary cytokine concentrations (IL-6, before and after surgery) between the patients with and without any treatments. Apparently, if validated with a larger number of patients enrolled prospectively, the analysis of the dynamic behavior of that cytokine level in the post-treatment phase might be an effective tool for the early identification/prediction of recurrence.

According to these observations, we also expect an appropriate combination of biomarkers to be validated in longitudinal studies, and their reliability was confirmed with respect to potential confounding factors such as behavioral habits and periodontitis.

In recent times, several reviews, which summarize and analyze the evidence of salivary cytokines as potential OSCC biomarkers, have been published [26,60,61]. The most distinguishing aspects that differentiate the present review from those works are: (1) the inclusion of longitudinal prospective studies, in which subjects are followed over time with repeated cytokine level monitoring; (2) the inclusion of studies investigating the cytokine level discriminatory power to distinguish early from advanced OSCC stages; and (3) the inclusion of several studies that use multiplex immunoassays to simultaneously quantify several salivary cytokines and chemokines.

Based on the present analysis, we conclude that a salivary screening test, built on a selected cytokine, seems quite realistic. The possibility of easy saliva sampling at the dentist’s chair will be the key step for advancing the diagnosis of oral cancer to an earlier stage, as outlined in Figure 2.

The major obstacle to this perspective arises from the observation that, in the analyzed studies, there is a wide variation of the average levels of salivary cytokines in both oncologic patients and healthy controls (Table 1, Table 2 and Table 3). Indeed, only the longitudinal approach of study Group III (Table 3) excluded the potential bias associated with inter–individual variability in cytokine levels.

Thus, further studies to confirm the reliability of a salivary screening test based on the quantification of selected cytokines are necessary. The improvement of such a test could be achieved by: (1) increasing population size at the multicenter level; (2) a broader representation of disease sites and stages of OSCC and OPMD; (3) single-analyte assays or low-to-mid-plex procedures, combining various cytokines; (4) standardization procedure of the cytokine quantitation, which envisages the normalization of cytokine concentration to total salivary protein content and the use of reagents meeting specific quality requirements for clinical laboratories; (5) control subjects matched to patients according to gingival condition.

## 4. Methods

### 4.1. Database Sources and Search Strategy

We searched the Scopus, Medline and WoS databases (Figure 1) for papers published in English between January 2005 and September 2019, to find potentially eligible studies.

The following search terms, “salivary cytokines”, “saliva cytokines”, “oral squamous cell carcinoma”, “oral cancer” and “salivary interleukins,” were used in various combinations, using “and” as a Boolean operator to combine concepts and narrow the search. We imported the results of each search into the EndNote software (Clarivate Analytics) for reference management, combining all records in a single group. The selection process adhered to the guidelines of the Preferred Reporting Items for Systematic Reviews and Meta-Analyses (PRISMA) statement [62]. Only five studies [27,35,39,40,49] potentially eligible for the final inclusion were retrieved from the reference lists of the accessed scientific literature.

Two reviewers, with either a biochemical or a clinical background, performed the eligibility assessment, without final disagreement. Full texts of all the included studies were obtained and thoroughly examined. Data relevant to the review question were extracted and documented in specifically designed spreadsheets. These data are summarized in Table 1, Table 2 and Table 3.

### 4.2. Criteria for Study Eligibility

Observational studies reporting on the alteration of any cytokine salivary concentration in patients with OSCC were considered suitable for the present review. In particular, the inclusion criteria for study eligibility were: (1) the evaluation of the salivary concentration of cytokines in patients with a histological diagnosis of OSCC; (2) comparison of salivary cytokine concentrations of OSCC patients with a control group; (3) comparison of pre-operative with post-operative salivary cytokine concentrations of OSCC patients, or comparison of post-operative salivary cytokine concentrations measured at different times during the follow-up period; 4) statistical analysis for a comparison of the salivary levels of cytokines. The main findings extracted from each study were the comparisons between the salivary levels of cytokines in healthy and pathological conditions, including the *p*-value, when significant (*p* ≤ 0.05).

The studies were assessed for specific risk of bias using the checklist obtained from the Joanna Briggs Institute [45]. The reviewers assessed the risk of bias, which were classified as no risk or low risk of bias (Appendix A). The demographic profile of OSCC patients involved in the included studies is presented in Appendix A.

Excluded studies were: (1) studies on OSCC patients under any pharmacological treatment or radiotherapy, because of potential interfering effects; (2) case reports, technological note articles, review articles; reports published in books; non-human studies; and non-English texts.

## Figures and Tables

**Figure 1 ijms-22-06795-f001:**
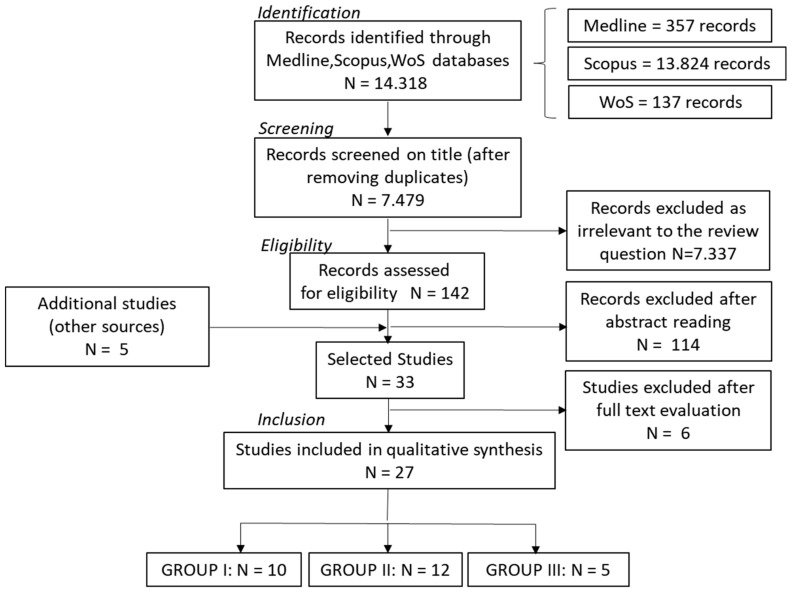
Flow diagram of the different phases of the systematic search and review. Phases are presented in accordance with the PRISMA statement. Group I and Group II articles include cross-sectional studies, while Group III includes longitudinal studies. Group I articles compare salivary cytokine levels of OSCC patients with control subjects; Group II articles include, together with the OSCC/control comparison, the comparison between salivary cytokine levels of different OSCC histological grades and clinical stages; Group III articles compare salivary cytokine levels before and after tumor excision treatment.

**Figure 2 ijms-22-06795-f002:**
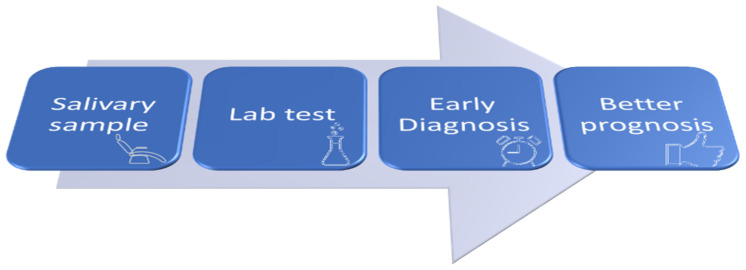
From the dentist chair, through the point-of-care analysis to the improvement of prognosis. Few steps for an early diagnosis and better survival of OSCC patients.

**Table 1 ijms-22-06795-t001:** GROUP I articles: evidence from the literature for salivary cytokines as candidate OSCC biomarkers.

Ref.	OCSS	Control	OPMD	Cytokine	OSCC vs. Control	*p*-Value	OPMD vs. Control	*p*-Value
[29]	28	31	29	IL-1β	OSCC > Control and OPMD	≤0.05	OPMD < Control	≤0.05
			^c^	IL-6	OSCC > Control and OPMD	<0.05	OPMD > Control	n. s.
				TNF-α	OSCC < Control > OPMD	n. s.	-	-
[32]	35	35	35	IL-6	n.p.		n.p.	
			^a–d^	IL-8	OSCC > Control	<0.0001	OPMD > Control	0.001
[33]	19	20	-	IL-1β	OSCC > Control	<0.05	-	-
				IL-6	OSCC > Control	<0.05	-	-
				IL-8	OSCC > Control	n. s.	-	-
				osteopontin	OSCC > Control	n. s.	-	-
[34]	13	13	13	IL-1	OSCC > Control	<0.01	OPMD > Control	<0.05
			^e^	IL-6	OSCC > Control	<0.01	OPMD > Control	<0.01
				IL-8	OSCC > Control	<0.01	OPMD > Control	<0.05
				TNF-α	OSCC > Control	<0.01	OPMD > Control	<0.05
[35]	19	19	19	IL-6	OSCC > Control and OPMD	<0.05	OPMD > Control	<0.05
			^b^	TNF-α	OSCC > Control and OPMD	<0.05	OPMD > Control	<0.05
[36]	18	56	-	IL-1α	OSCC > Control	<0.0001	-	-
				IL-6	OSCC > Control	<0.0001	-	-
				IL-8	OSCC > Control	<0.0001	-	-
				TNF-α	OSCC > Control	<0.0001	-	-
				VEGF-a	OSCC > Control	<0.0001	-	-
[37]	30	20	-	IL-1α	OSCC > Control	<0.001	-	-
				IL-6	OSCC > Control	<0.05	-	-
				IL-8	OSCC > Control	<0.001	-	-
				GM-CSF	OSCC > Control	<0.05	-	-
[38]	9	9	-	IL-1α	OSCC > Control	n. s.	-	-
				IL-6	OSCC > Control	<0.05	-	-
				IL-8	OSCC > Control	n. s.	-	-
				TNF-α	OSCC > Control	n. s.	-	-
[39]	78	40		IL-10	OSCC > Control	0.00002	-	-
				TNF-α	OSCC > Control	0.00002	-	-
				TGF-β	OSCC > Control	0.00002	-	-
				VEGF	OSCC > Control	0.0000		
[40]	60	60	60	IL-1β	OSCC > Control	=0.01	-	-
					OSCC > OPMD	0.004	-	-
				IL-8	OSCC > Control	<0.0001	-	-
					OSCC > OPMD	<0.0001	-	-

OSCC, oral squamous cell carcinoma; OPMD, oral potentially malignant disorders; n. s., non-significant; n.p., not performed (cytokine level resulted undetectable in most of OPMD and CONTROL subjects). ^a^ oral sub mucous fibrosis, OMSF; ^b^ oral lichen planus, OLP; ^c^ Leukoplakia; ^d^ Eritroplakia; ^e^ epithelial dysplasia.

**Table 2 ijms-22-06795-t002:** GROUP II articles: evidence from the literature for salivary cytokines as candidate OSCC biomarkers.

Ref.	OSCC	Control	OPMD	Cytokine	OSCC vs. Control/OPMD	*p*-Value	OPMD vs. Control	*p*-Value	OSCC Histological Grade and/or Stages	*p*-Value
[28]	30	33	-	IL-10	OSCC > Control	0.004	-	-	WD > Control	0.001
				IL-13	OSCC > Control	0.01	-	-	-	n. s.
				IL-1RA	OSCC > Control	n. s.			PD > MD PD > WD PD > Control	0.000 0.002 0.000
				IL-4	OSCC > Control	n. s.	-	-	-	-
[41]	25	25	25 leucoplakia + OSMF	IL-8	OSCC > Control	<0.0001	OPMD > Control	n. s.	stage IV > stages II and III MD > WD	n. s n. s.
					OSCC > OPMD	<0.0001				
[42] *	41	24	-	IL-6	OSCC > Control	<0.001	-	-	stage III/IV > Control stage I/II > Control	<0.01 <0.01
				IL-8	OSCC > Control	0.001			stage III/IV > Control stage I/II > Control	<0.01 <0.05
				IL-1β	OSCC > Control	0.002			stage III/IV > Control stage I/II > Control	<0.01 <0.05
				TNF-α	OSCC > Control	0.001			stage III/IV > Control stage I/II > Control	<0.01 <0.05
				IFN-γ	OSCC > Control	0.036			stage III/IV > Control	<0.01
				MIP-1β	OSCC > Control	0.016			stage III/IV > Control stage I/II > Control	<0.05 <0.05
				Eotaxin	OSCC > Control	0.03			-	-
				GRO	OSCC > Control	n. s.			stage I/II > Control stage III/IV > Control	<0.05 <0.05
[43]	100	100	50 + 50 leucoplakia + OSMF	IL-6	OSCC > OPMD	<0.01	OPMD > Control	<0.05	PD > MD PD > WD MD > WD	<0.05 <0.01 <0.05
[44]	100	100	50 + 50 leucoplakia + OSMF	IL-8	OSCC > Control	<0.001	OPMD > Control	<0.05	PD > MD PD > WD MD > WD T1 > OPMD	<0.01 <0.001 <0.05 <0.05
					OSCC > OPMD	<0.01				
[45]	30	30	30 leucoplakia	TNF-α	OSCC > Control	<0.001	OPMD > Control	<0.001	MD/PD > WD stages III and IV > stages I and II	<0.001 <0.030
					OSCC > OPMD	<0.001				
[46]	100	100	50 + 50 leucoplakia + OSMF	TNF-α	OSCC > Control	<0.001	OPMD > Control	<0.05	PD > WD MD > WD stage IV > all other stages	<0.01 <0.05 <0.01
					OSCC > OPMD	<0.05				
[47]	30	30	30 leucoplakia	TNF-α	OSCC > Control	<0.01	OPMD > Control	<0.01	PD > MD, WD	<0.01 <0.01
					OSCC > OPMD	<0.01				
[48]	18	21	41 OLP	IL-6	OSCC > Control	<0.001	-	-	stage IV > Control stage I > Control	0.002 0.001
					OSCC > OPMD	≤0.001				
				IL-8	OSCC > Control	0.014				n. s.
[49]	35	51	-	IL-8	OSCC > Control	<0.0001	-	-	T1/T2 stage > Control T3/T4 stage > Control	0.004 <0.0001
				IL-1β	OSCC > Control	<0.0001			T1/T2 stage > Control T3/T4 stage > Control	0.0002 <0.0001
[50]	25	25	25 leucoplakia	IL-6	OSCC > Control	<0.001	OPMD > Control	<0.001	stage IV > stage II	0.021
					OSCC > OPMD	<0.001				
[27]	66	25	66 leucoplakia	IL-1α	OSCC > Control	n. s.	OPMD > Control	n. s.	T1/T2 stage > Control OPMD > T1/T2 stage T1/T2 stage> T3/T4 stage	n. s. n. s. n. s.
					OSCC > OPMD	n. s.				
				IL-6	OSCC > Control	≤0.0001	OPMD > Control	0.001	T1/T2 stage > Control T1/T2 stage > OPMD T3/T4 stage > T1/T2 stage	<0.001 <0.001 0.01
					OSCC > OPMD	≤0.0001				
				IL-8	OSCC > Control	≤0.0001	OPMD > Control	0.004	T1/T2 stage > Control T1/T2 stage > OPMD T3/T4 stage > T1/T2 stage	<0.001 0.05 n. s.
					OSCC > OPMD	≤0.0001				
				TNF-α	OSCC > Control	≤0.0001	OPMD > Control	0.001	T1/T2 stage > Control T1/T2 stage > OPMD T3/T4 stage > T1/T2 stage	<0.001 <0.001 0.01
					OSCC > OPMD	≤0.0001				
				HCC-1	OSCC > Control	≤0.0001	OPMD > Control	0.002	T1/T2 stage > Control T1/T2 stage > OPMD T3/T4 stage > T1/T2 stage	<0.001 0.01 n. s.
					OSCC > OPMD	≤0.0001				
				MCP-1	OSCC > Control	≤0.01	OPMD > Control	0.001	T1/T2 stage > Control T1/T2 stage < OPMD T3/T4 stage < T1/T2 stage	0.003 n. s. n. s.
					OSCC > OPMD	n. s.				
				PF-4	OSCC > Control	≤0.002	OPMD > Control	n. s.	T1/T2 stage > Control T1/T2 stage > OPMD T3/T4 stage > T1/T2 stage	0.01 <0.001 n. s.
					OSCC > OPMD	≤0.0001				

OSCC, oral squamous cell carcinoma; OPMD, oral potentially malignant disorders; OMSF, oral submucous fibrosis; n. s., non-significant. WD, MD and PD correspond, respectively, to Well Differentiated, Moderately Differentiated and Poorly Differentiated histological grades; Stages I-IV correspond to the four OSCC stage groups; T1-T4 stages correspond to tumor different sizes in Tumor-Node-Metastasis (TNM) staging system. * This study analyses 14 cytokines and here are reported only the significant variations.

**Table 3 ijms-22-06795-t003:** GROUP III articles: evidence from the literature for salivary cytokines as candidate OSCC biomarkers.

Ref.	OSCC	Control	Cytokine	OSCC vs. Control	*p*-Value	Pre/Post- Operative	*p*-Value	Post-Operative	*p*-Value	24 Months after Surgery	*p*-Value
[51]	25	25	IL-17	OSCC > Control	<0.001	Pre > Post ^b^	<0.001	-	-	-	-
[52] ^ç^	20	-	IL-8	-		Pre > Post ^c^	0.004	-	-	-	-
			IL-6			Pre > Post ^c^	0.005	-	-	-	-
			VEGF			Pre > Post ^c^	0.014	-	-	-	-
			MIP-1β			Pre > Post ^c^	0.033	-	-	-	-
			IP-10			Pre > Post ^c^	0.047	-	-	-	-
			IL-1β			Pre > Post ^c^	0.049	-	-	-	-
			INF-γ			Pre < Post ^c^	0.036	-	-	-	-
			IL-5			Pre < Post ^c^	0.048	-	-	-	-
[53]	16	-	IL-1β	-		Pre > Post ^d^	<0.05	-	-	-	-
[54]	27	21	IL-6	OSCC > Control	0.002	Pre > Post ^a^	n. s.	Early recurrence ^ (+) > recurrence (−)	0.02	-	-
[55] ^§^	27	21	IL-6	-						Post-op. > 24 mos. after surgery	0.006
										24 mos. after surgery > Control	n. s.
										Late recurrence * (+) > recurrence (−)	0.03

OSCC, oral squamous cell carcinoma; n. s., non-significant; mos., months; pre/post-operative, comparison of cytokine pre- and post-operative salivary levels; Post-operative, comparison of cytokine post-operative salivary levels; 24 months after surgery, comparison of cytokine salivary levels measured 24 months after surgery. ^Ç^ This study analyzes 27 cytokines and here are reported only the significant variations. ^a–d^ Interval between before- and after-surgery saliva collections was 30 ± 18 days, 12 days, 2 months and 1.5 months for a, b, c, d, respectively. ^§^ This study is an extension of Sato et al., 2013 study [54], evaluating post-operative IL-6 concentration of the patients already enrolled in the previous study. ^ Early loco regional recurrence occurred within 24 months after surgery; * late loco regional recurrence occurred in the 24–48 months after surgery; (−) subjects without recurrence.

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
