# Peer review of "Salivary Cytokines as Biomarkers for Oral Squamous Cell Carcinoma: A Systematic Review"

_ijms, 2021, doi:10.3390/ijms22136795_

Round 1

Reviewer 1 Report

In this review, authors showed that salivary IL-6, IL-8 and TNF-α levels are significantly high in the patient with OSCC compared to OPMD patients and healthy controls. These salivary cytokines may be useful biomarkers for detecting early OSCC. This review is logical and interesting. The reviewer feels that this review is suitable for publication in the International Journal of Molecular Sciences.

Author Response

We appreciate the time and efforts by the Reviewers and Editors in reviewing this manuscript.

We welcomed the request of Reviewer #2 and introduced Table 2S in the Supplementary Material.

New text is highlighted in yellow (Methods section).

Sincerely yours,

Prof. Thelma Pertinhez

Reviewer 2 Report

In the present manuscript Ferrari E. and collaborators show an observational study to demonstrate evidence to support the role of specific cytokines in diagnosis and/or progression of OSCC.

The review thoroughly analyzes 27 papers obtained from reference bibliographic scientific/medicine libraries and databases.

The focus of the different studies is to compare cytokine levels between healthy/OPMD/OSCC (TNM stage and surgical resection) patients to correlate the cytokine expression with disease progression in order to assess the viability of such markers in the diagnosis of the OSCC disease and, importantly, in the early diagnosis.

The study is well presented and includes many relevant data that could be of potential interest for scientific and clinical community. The main challenges in the use of salivary CKs as biomarkers are relative to standardization methods to obtain valid references indexes are also discussed.

This is a quite descriptive manuscript. For an optimal publication I would suggest to include in the manuscript the data about the cohort used in each of the studies (population origin/sex/age) given that that data could also affect the expression pattern of salivary cytokines and it would give wider vision about potential data extrapolation.

Author Response

(The authors gave the same response as above.)
